# Fabrication and Characterization of Ferrofluidic-Based Wire-Wound and Wire-Bonded Type Inductor for Continuous RF Tunable Inductor

**Fatemeh Bani Torfian Hoveizavi [1,2], Nuha Abdul Rhaffor [1], Sofiyah Sal Hamid [1], Khairu Anuar Mohamed Zain [1], Shukri Korakkottil Kunhi Mohd [1], Mohd Tafir Mustaffa [2]** and **Asrulnizam Abd Manaf [1,2,*]**

[1] Collaborative Microelectronic Design Excellence Center, Universiti Sains Malaysia, Sains@USM, Level 1, Block C, No. 10, Persiaran Bukit Jambul, Bayan Lepas 11900, P. Pinang, Malaysia; banitorfian.f@gmail.com (F.B.T.H.); nuha@usm.my (N.A.R.); sofiyah@usm.my (S.S.H.); anuar@usm.my (K.A.M.Z.); shukri.mohd@usm.my (S.K.K.M.)

[2] School of Electrical and Electronic Engineering, Engineering Campus, Universiti Sains Malaysia, Nibong Tebal 14300, P. Pinang, Malaysia; tafir@usm.my

\* Correspondence: eeasrulnizam@usm.my; Tel.: +604-6535709



**Featured Application: The growing market of Internet of Things (IOT) wireless multi-band systems creates the demand for high-quality and highly linear variable RF components. These components, including discrete and microelectronic mechanical systems (MEMS) inductors, have a wide range of applications in RF (radio frequency) transceivers, such as voltage-controlled oscillator (VCO), low-noise amplifier (LNA), direct current—direct current (DC–DC) converters, matching networks, multi-band filters, multi-band RF circuits, RF power amplifiers in radio transmitters, high-isolated switches, and reconfigurable antennas.**

**Abstract:** This paper reports on the novel implementation of a tunable solenoid inductor using the fluid-based inductance varying technique. The concept utilized material permeability variation that directly modifies self-induced magnetic flow density inside the coil, which in turn creates a variation of the inductance value. The core is formed by a channel which allows the circulation of a liquid through it. The liquid proposed for this technique has ferromagnetic behavior, called ferrofluid, with a magnetic permeability higher than unity. To evaluate the proposed technique, two different types of solenoid inductor were designed, simulated and measured. The two structures are wire-wound and wire bond solenoid inductors. The structures are simulated in a 3D EM analysis tool followed by fabrication, test and measurement. The wire-bonded-based inductor showed a quality factor of 12.7 at 310 MHz, with 81% tuning ratio, by using ferrofluid EMG 901. The wire-wound-based inductor showed that the maximum tuning ratio is 90.6% with quality factor 31.3 at 300 MHz for ferro fluidic EMG 901. The maximum measured tuning ranges were equal to 83.5% and 56.2% for the wire-wound type and the wire-bonded one, respectively. The measurement results for the proposed technique showed a very high tuning range, as well as high quality factor and continuous tunability.

**Keywords:** tunable solenoid inductor; fluid-based inductance; wire-wound inductor; wire bond inductors

## 1. Introduction

With the evolution of wireless communications, especially market demand for a smaller communication device, there is a tremendous interest and need for integrated, less costly and

more efficient electronic components. Tunable capacitors were the only known technology that could utilize these incredible market chances. Recently, the demand for the tunability feature of the inductor increased, especially in radio frequency (RF) application, to create standard reconfigurable wireless systems utilizing single miniaturized on-chip solution. This includes discrete and micro electro-mechanical system (MEMS) inductors that have a wide range of applications in RF transceiver block, such as voltage controlled oscillator (VCO) [1,2], low-noise amplifier (LNA) [3], wireless matching network [4–7], multi-band filters [8], multi-band RF circuits [9], RF power amplifier [10], and reconfigurable antenna [11–16]. Beside using the tunable inductor for these RF applications, the presence of its variable functions can compensate a temperature or frequency drift, system performance due to process variation and malfunction because of component aging. However, research to create such a component is still at early stages, since obtaining high inductance tuning range and high quality factor simultaneously is very challenging.

Previous work has proposed various discrete inductor tuning techniques, such as switched turns [16], switched mutual inductance [17], bimorph-effect-based varied-coupling [18] and switched magnetic field [19]. Unfortunately, narrow tuning range observed plus the obtained quality factor is quite low. Later, the research in this field has moved to MEMS technique, where tunable inductors are sometimes based on the self-assembling variable inductor, the transformer-type interleaved spiral or stacked coils inductor [20,21]. Although the fabrication for this type of MEMS is cost effective due to single layer metal utilization, the obtained quality factor is low because of skin effect. As for other types of MEMS tunable technique, which are switch-controlled inductor [22], thermal actuator that controls the spacing between main and secondary coils [23], and inductor tuned by the means of bimorph actuation [24], these mechanisms suffer from a few constraints, such as discrete-step tuning capability, lower tuning range, and high fabrication cost. The goal of this work is to design a novel tunable inductor with high tuning range and high-quality factor for radio frequency applications using microfluids. It is achieved by manipulating and the precise control of fluid behavior, which result changes to the inductor characteristic, thus, changing the inductance value. In this work, Section 2 describes principle of inductor theory. Design, implementation and measurement results are presented in Sections 3–5, respectively. Finally, a conclusion of the work is given in Section 5.

## 2. Implementation of Tunable Microfluidic Inductor

### 2.1. Wire-Wound Inductor

A prototype of the wire-wound inductor was fabricated on FR-4 Printed Circuitry Board (PCB). Figure 1 shows an image of the implemented wire-wound inductor. Insulated copper wire was used as the coil because of its high electrical conductivity to improve the quality factor and its flexibility to easily formed into coil. In addition, micropipe and polydimethylsiloxane (PDMS) were used to realize the inner channel. The designed wire-wound inductors have 10 turns and 1-mm inner diameter with 0.5 mm wire thickness, as shown in Table 1. The solenoid core is realized by a 1-mm micropipe. Figure 1a shows the micropipe used to create microchannel mold, while Figure 1b,c show dimension of the device and the completed wire-wound inductor with microchannel PDMS, respectively.

**Table 1.** Dimension of fabricated wire-wound inductor structure.

| Type | Wire Thickness | Number of Turns | Inner Diameter |
|------|----------------|-----------------|----------------|
| Wire-wound solenoid | 0.5 mm | 10 | 1 mm |

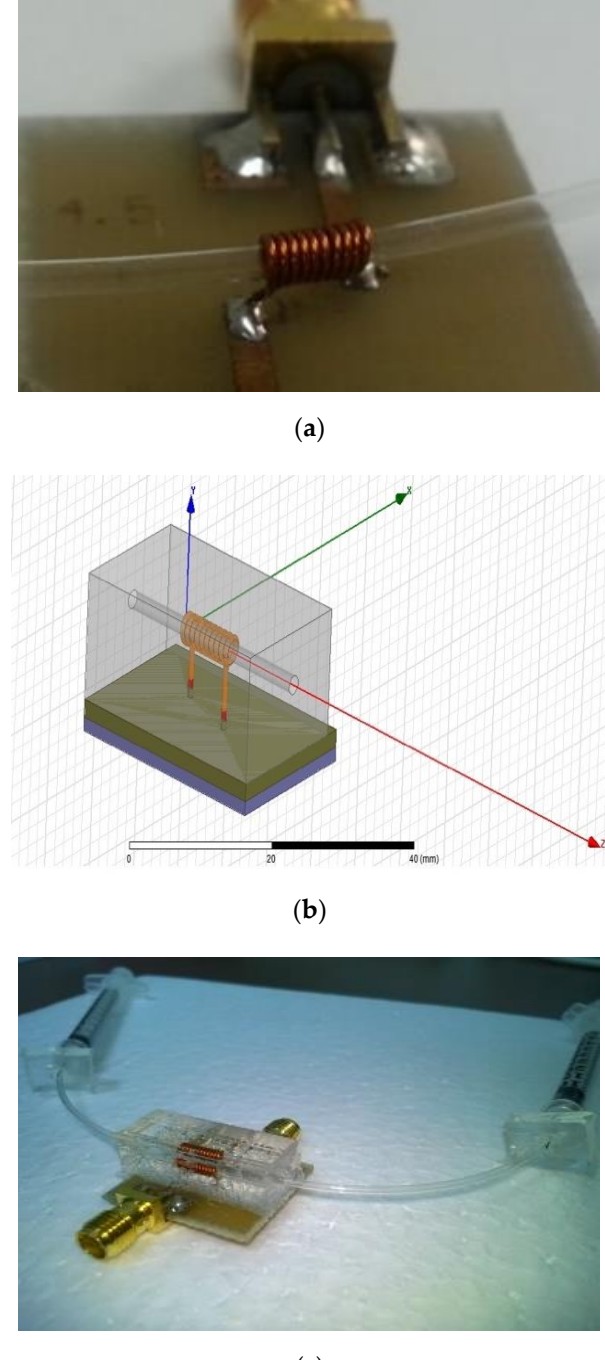

**Figure 1.** (**a**) Fabricated wire-wound inductor prototype with micro-pipe, and (**b**) Dimension of complete wire wound structure (**c**) photograph image of complete wire-wound inductor [25].

*2.2. Wire Bond Inductor*

The implemented wire bond inductor is as shown in Figure 2. Coil structure was formed by two components; copper traces on the PCB and bonding wire as demonstrate in Figure 2a. The PCB used in this work is ROGERS4003, with 0.8-mm dielectric thickness, relative permittivity of 3.38, loss tangent of 0.0027, metal conductor thickness of 18μm and metal conductivity of $5.8 \times 10^{7}$ S/m. The metal width at the base is 250 μm with a space of 250 μm. The bond wire is gold with 25 μm diameter.

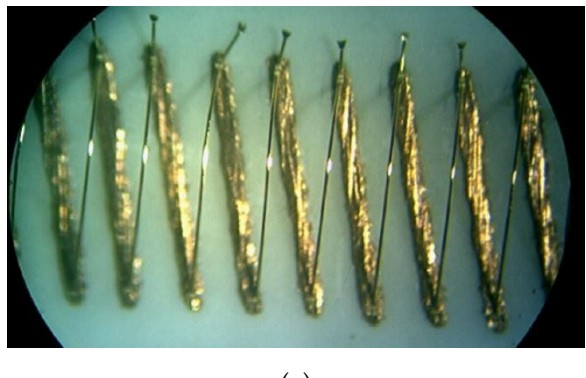

(**a**)

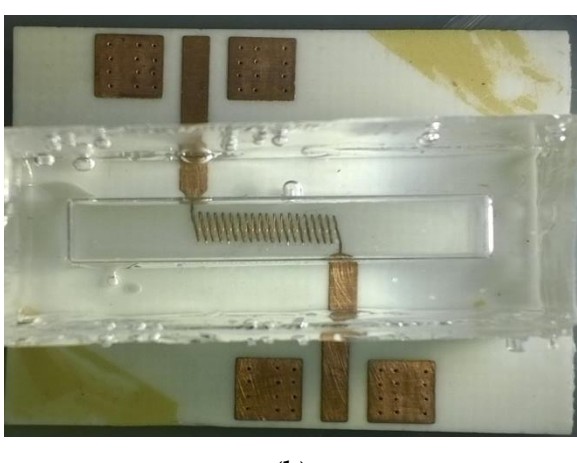

(**b**)

**Figure 2.** Implemented tunable wire bond inductor. (**a**) The micrograph of the fabricated wire bond solenoid inductor. (**b**) The micrograph of the fabricated wire bond solenoid inductor with Polydimethylsiloxane (PDMS)-based micro channel.

As for catalyst to change the substrate property, thus changing the inductance value, PDMS based microchannel was attached onto the PCB, as shown in Figure 2b. The specification of wire bond solenoid inductor for metal thickness, number of turns, space between turns and bond diameter were 250 μm, 10, 250 μm and 25 μm, respectively.

## 3. Measurement Setup and Ferrofluidic Properties

### 3.1. Characterization of Tunable Inductor

The fabricated wound and bonded structure inductor contains two ports to connect to the network analyzer, with sub-miniature version A (SMA) connectors. The network analyzer was calibrated up to the end of its ports' connectors (up to the ports of the device under test), so that the insertion and mismatch loss due to the ports and their corresponding connectors are eliminated. The calibration kit used for this work is SOLT, which is short, open, load and through. The wire-wound solenoid inductor was fabricated using Rogers 4003 laminate.

In addition, to remove the connectors and lines effect of the PCB until the heads of the inductor, a SOLT calibration kit was designed and fabricated using FR-4 PCB. The FR-4 laminate is the identical material used for fabricated spiral inductor. The calibration kit fabricated for this work is shown in Figure 3.

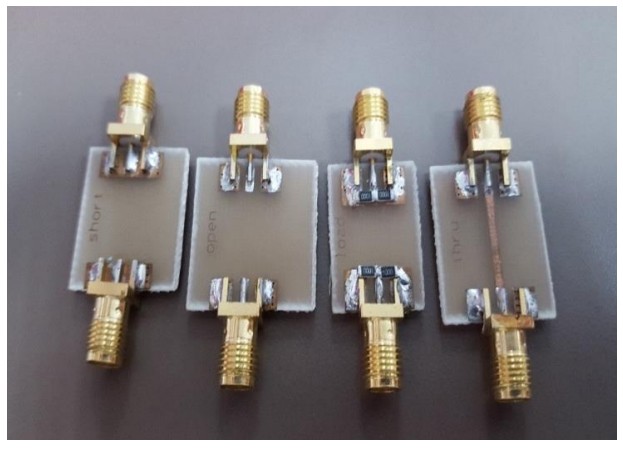

(**a**)

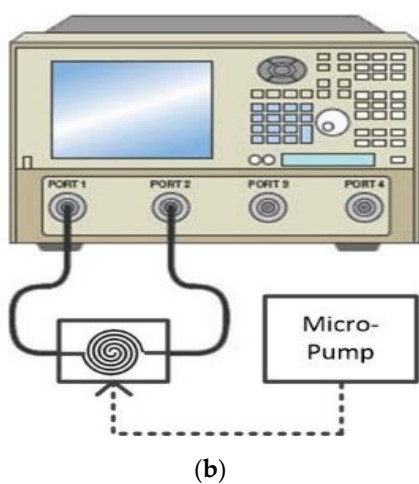

(**b**)

**Figure 3.** (**a**) The calibration kit for inductor characterization. (**b**) The measurement setup for inductor characterization.

The inductor was measured in an empty state and the state when the ferrofluid was injected to the channel. The measured S-parameter data were used to determine the inductance and quality factor. The measurement setup, including the network analyzer and injection tools used for measuring inductance and quality factors, is shown in Figure 3. The wire-wound solenoid inductor was first measured when its channel was empty. Then, the ferrofluids, such as EMG901, were injected into the channel gradually until it was completely filled. This causes the inductance to increase continuously. The maximum inductance is achieved when the channel is fully injected. The same procedure was done for another three ferrofluids with different permeability and magnetic saturation. This method was implemented for both structures.

*3.2. Ferrofluidic Properties*

Ferrofluid is a type of material that can act like a liquid and a magnetic solid. It is made of nanoscale ferromagnetic particles and suspended in a liquid (such as oil). When no magnet is nearby, the material acts like a liquid, but when a magnet is nearby, the particles become temporarily magnetized and the ferrofluid acts like a solid. When the magnet is removed, it returns to its liquid state. The particles, which have an average size of about 100 Å (10 nm), are coated with a stabilizing dispersing agent (surfactant) which prevents particle agglomeration, even when a strong magnetic field gradient is applied to the ferrofluid. In this work, the ferrofluid was supplied from Ferrotec Co. Ltd. (www.ferrotec.com). As shown in Table 2, in this work, four ferrofluids, referred to as EMG901,

EMG905, EMG909 and EMG911, were applied to, respectively, initial magnetic permeabilities of 5.4, 3.1, 1.9 and 1.3; magnetic saturation equal to 66 mT, 44 mT, 22 mT and 11 mT, and magnetic particle concentration 11.8%, 7.8%, 3.9% and 2%.

**Table 2.** Properties of ferrofluid.

| Ferrofluidic Type | Magnetic Permeability | Magnetic Saturation | Particle Concentration |
|---|---|---|---|
| EMG901 | 5.4 | 66 mT | 11.8% |
| EMG905 | 3.1 | 44 mT | 7.8% |
| EMG909 | 1.9 | 22 mT | 3.9% |
| EMG911 | 1.3 | 11 mT | 2.0% |

## 4. Results and Discussion

The wire bond solenoid inductor is modeled and simulated in High Frequency Structure Simulator (HFSS) as shown in Figure 4a. As can be seen, the metal base is implemented on a ROGERS4003 dielectric. It is recommended to use a relative permittivity of 3.55 for simulation. The wire bonds are modeled, as well, to form the solenoid inductor. Two excitation ports act as the terminals with a common reference ground. A three-dimensional solenoid inductor for both inductor types was designed in the HFSS EM simulation tool. Figure 4b shows the bond wire (bridge) connections on the base part. The PCB used in this work is ROGERS4003, with 0.8-mm dielectric thickness, a relative permittivity of 3.38, a loss tangent of 0.0027, a metal conductor thickness of 18 μm and a metal conductivity of $5.8 \times 10^7$ S/m. The metal width at the base is 10 mL with a space of 10mL. The bond wire is gold (Au) with 25 μm diameter. Ferrofluid was injected into the channel, then it remains static in channel during measurement. The characterization of the inductance was performed based on the percentage of liquid in the channel. As explained in Section 3.2, ferrofluid was modeled as solid magnetic bar when the particles were magnetized.

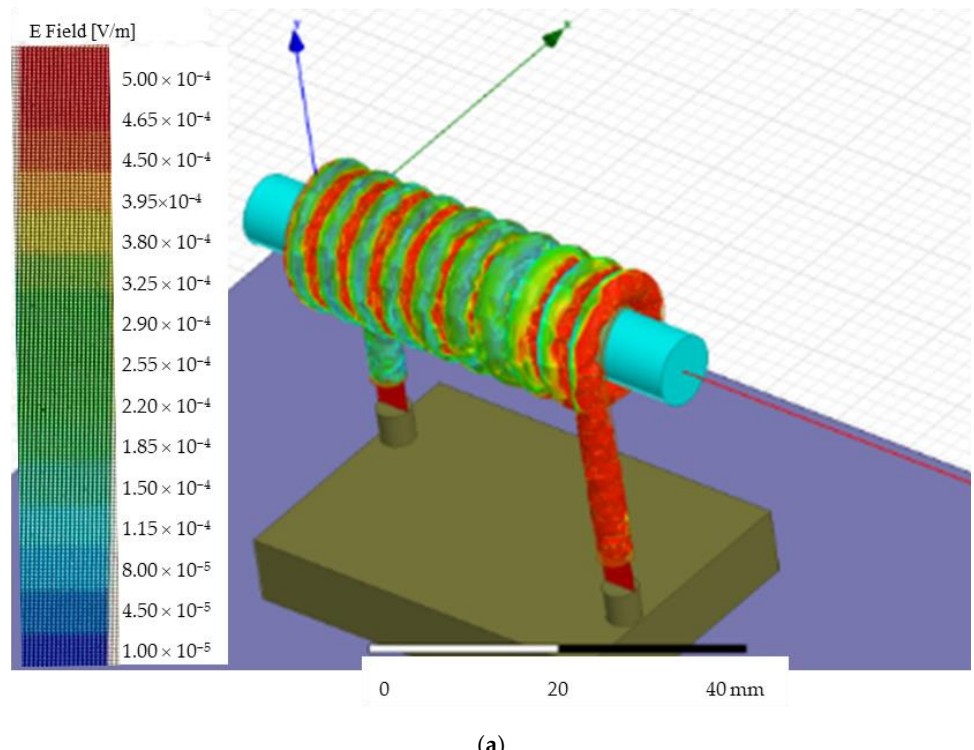

(a)

**Figure 4.** *Cont.*

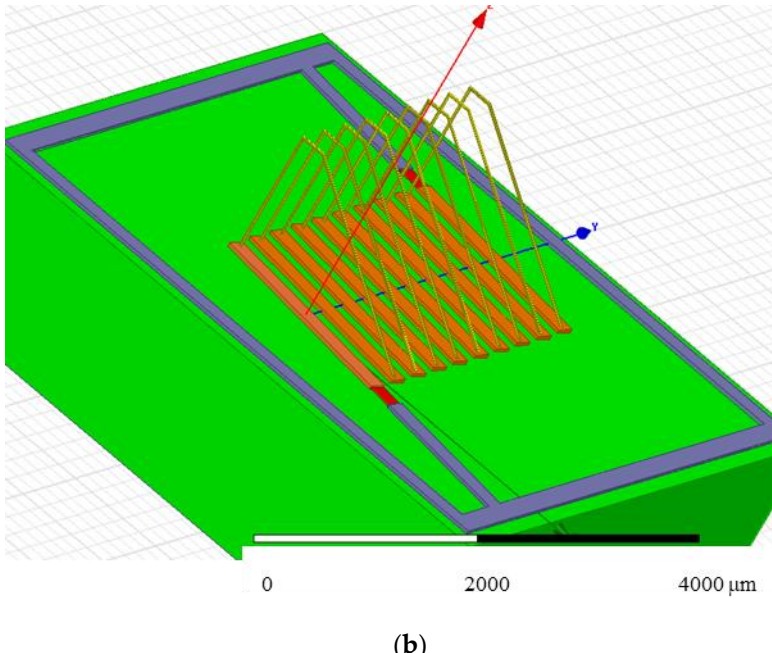

(**b**)

**Figure 4.** The 3D view of inductor designed in High Frequency Structure Simulator (HFSS) (**a**) Wire-wound inductor [25], (**b**) wire bond inductor [26].

Using the simulated data, the discrepancy between measurement results and the ideal solution can be obtained. Figure 4a,b show the simulated 3D view of the wire-wound inductor and wire bond inductor, respectively.

### 4.1. Wire-Wound Solenoid Inductor

Figure 5a illustrates the simulation results of inductance variation, corresponding to the full injection of the different ferrofluids. As can be seen in this figure, the minimum inductance is obtained at vacuum. When the channel is empty, the core is filled by air, where the permeability is approximately equal to vacuum. As tunable fluidic based inductors, ferrofluids will be used for injection into the core. According to the permeability of the injected liquid, the inductance is varied. In this work, four ferrofluids, referred to as EMG901, EMG905, EMG909 and EMG911, were applied to, respectively, initial magnetic permeabilities of 5.4, 3.1, 1.9 and 1.3, magnetic saturation equal to 66 mT, 44 mT, 22 mT and 11 mT, and magnetic particle concentrations 11.8%, 7.8%, 3.9% and 2%.

The maximum inductance is achieved for EMG901. EMG901 has the highest permeability among the applied ferrofluids. The simulated inductance for vacuum is 54.4 nH at 100 MHz, and increases to 65.4, 86.1, 105.6 and 187.3 nH for EMG911, EMG909, EMG905 and EMG901, respectively. Figure 5b shows the simulated corresponding quality factor. For all fluids, the quality factor obtained is over 300 at 100 MHz. The quality factor slightly decreases when the ferrofluids are injected. However, the quality factor reduction is more severe for EMG901. Figure 5c shows the simulated tuning ratio. The maximum tuning ratio obtained is larger than 243% for EMG901 at 100 MHz. For other ferrofluids, the tuning ratio obtained is from 20.22% to 243%, for a frequency range of a few MHz to 900 MHz.

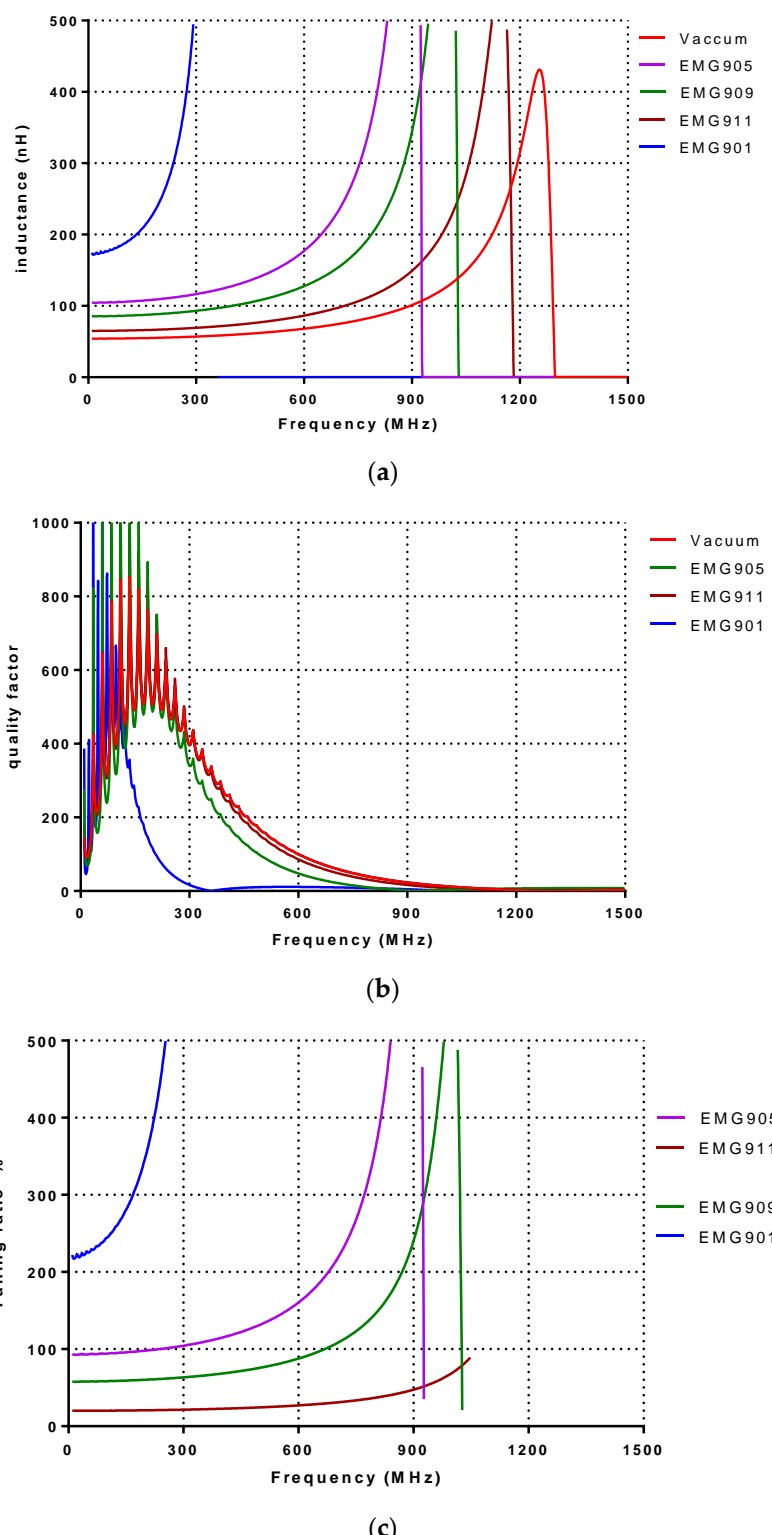

**Figure 5.** Simulated results of wire-wound inductor. (**a**) Simulated inductance, (**b**) Quality factor, and (**c**) Tuning ratio.

Figure 6 shows the measurement result of the fabricated wire-wound inductor. As simulated, the maximum inductance is obtained for EMG901, while the vacuum presents the minimum achievable inductance. In Figure 6a, the measured inductance values are 55, 62.4, 68.8, 78.6 and 101 nH at 200 MHz, respectively, for vacuum, EMG911, EMG909, EMG905 and EMG901. The maximum quality factor is

achieved during vacuum at 120 for 150 MHz. The quality factor slightly drops when the ferrofluids are injected. Figure 6c shows the tuning ratio of the fabricated wire-wound inductor. The measured tuning ratio for EMG901 at 200 MHz is 83.5%, while it increases to 90.6% at 300 MHz. As for the EMG911, EMG 909 and EMG905, the tuning ratios are 13.3%, 25% and 42.9%, respectively. Table 3 showed the comparison between the simulated and measurement result for the wire-wound inductor. As shown in the Table 3, the Q factor and tuning range for the simulated and measurement result at 100 MHz for EMG901 were 300%, 243%, 53.7% and 83.5%, respectively. The reason for the large difference between simulation and measurement data is because, in simulation, one only considers open space without close channel, thus, the effect of inner wall surface properties such as surface tensioning and meniscus force were not considered in the simulation model. Thus, the simulation was assumed to be a smooth movement of fluid in the channel. The simulation also assumes, as an ideal case, that parasitic capacitance is not included in this model. In the measurement result, the difficulties of movement fluidic and parasitic capacitance were affecting the value of inductance, and this contributed to the low tuning ratio and Q factor. The tuning ratio percentage is calculated by

$$\left(\frac{L_2 - L_1}{L_1}\right) \times 100 \tag{1}$$

where $L_1$ and $L_2$ are the minimum and maximum achieved inductances.

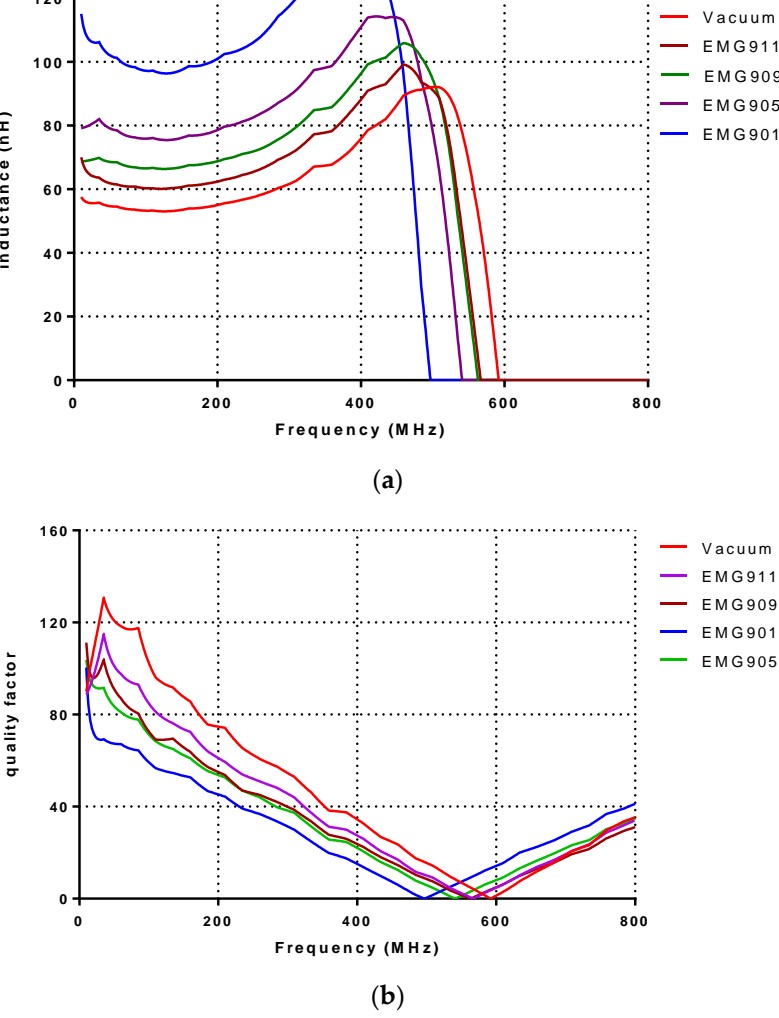

(**a**)

(**b**)

**Figure 6.** *Cont.*

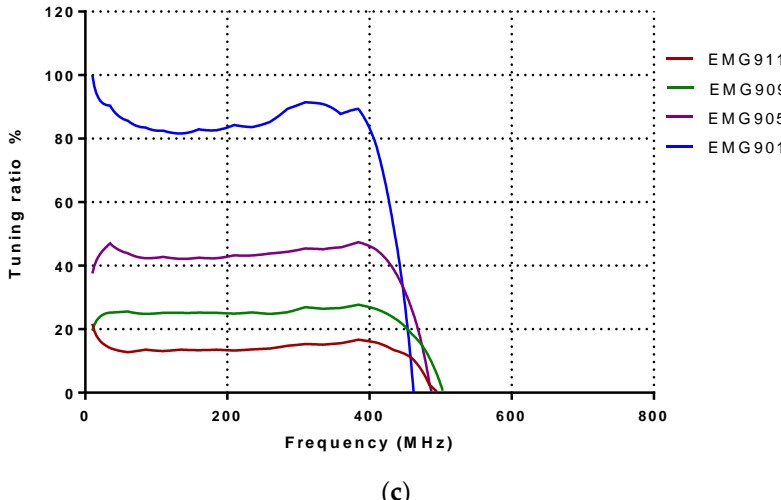

(**c**)

**Figure 6.** Measurement results of wire-wound inductor. (**a**) Measured inductance, (**b**) Quality factor, and (**c**) Tuning ratio.

**Table 3.** Summary of characterization of wire-wound structure tunable inductor.

| Wire-Wound @100MHz | Sim. L | Sim. Q | Meas. L | Meas. Q | Sim. TR% | Meas. TR% |
|---|---|---|---|---|---|---|
| **Vacuum** | 54.4 | 300 | 55 | 200 | - | - |
| EMG901 | 187.3 | 300 | 101 | 53.7 | 243% | 83.5% |
| EMG905 | 105.6 | 300 | 78.6 | 55 | 94.11% | 42.5% |
| EMG909 | 86.1 | 300 | 68.8 | 61.1 | 58.27% | 25% |
| EMG911 | 65.4 | 300 | 62.4 | 74.4 | 20.22% | 13.3% |

*4.2. Wire Bond Solenoid Inductor*

Figure 7a shows the simulated result of the wire bond inductor that varies from 47 to 181 at 200 MHz. For vacuum, EMG911, EMG909, EMG905 and EMG901, the simulated inductances are 47.8, 61.5, 85.5, 107.8 and 286.3 nH at 300 MHz, respectively. The contra result observed is compared to the wire-wound inductor, where, according to Figure 7b, the maximum quality factor obtained is 70 at 350 MHz for EMG901. Figure 7c shows the simulated tuning ratio, where the maximum tuning ratio belongs to the EMG901, equal to 498% at 300 MHz. According to the inductance variations, the minimum and the maximum tuning ratio correspond to EMG911 and EMG901, respectively.

Figure 8 shows the measurement result for the wire bond inductor. As shown in Figure 7a, a minimum inductance value of 42.6 nH at 150 MHz showed at vacuum. The maximum inductance obtained is 66.6 nH at 150 MHz, with EMG901 ferrofluids. The measured inductances for EMG911, EMG909 and EMG905 are 46.2, 49.8, 54.2 nH at 150 MHz, respectively. The measured quality factors are shown in Figure 8b. The maximum quality factor obtained with a vacuum at 39.1 for 150 MHz, while EMG901-filled channel presents the minimum achieved quality factor of 17.2 at 150 MHz. Figure 8c shows the tuning ratio. The maximum achieved tuning ratio is 81% for EMG901 at 310 MHz. Table 2 showed the summary of performance for the wire-bonded structure tunable inductor. Table 4 shows the characterization of wire-bonded structure tunable inductor with different type of ferrofluid. Table 5 shows the summary of the fabricated tunable inductor compared to the other work using solenoid structure technology. As shown in Table 5, the fabricated tunable inductor proved to be a promising and alternative tunable inductor.

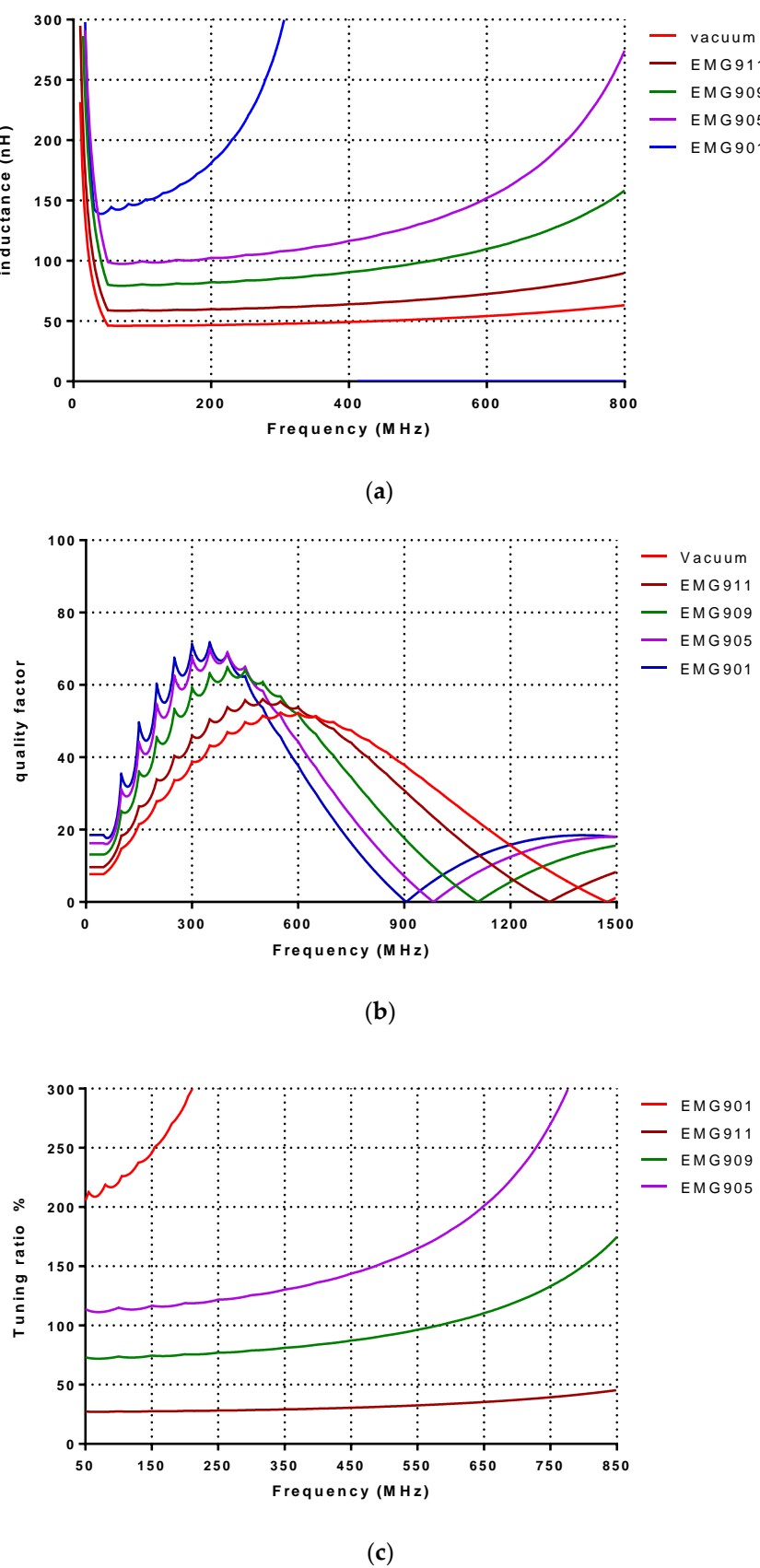

**Figure 7.** Simulation results of wire bond inductor. (**a**) Simulated inductance, (**b**) Quality factor, and (**c**) Tuning ratio.

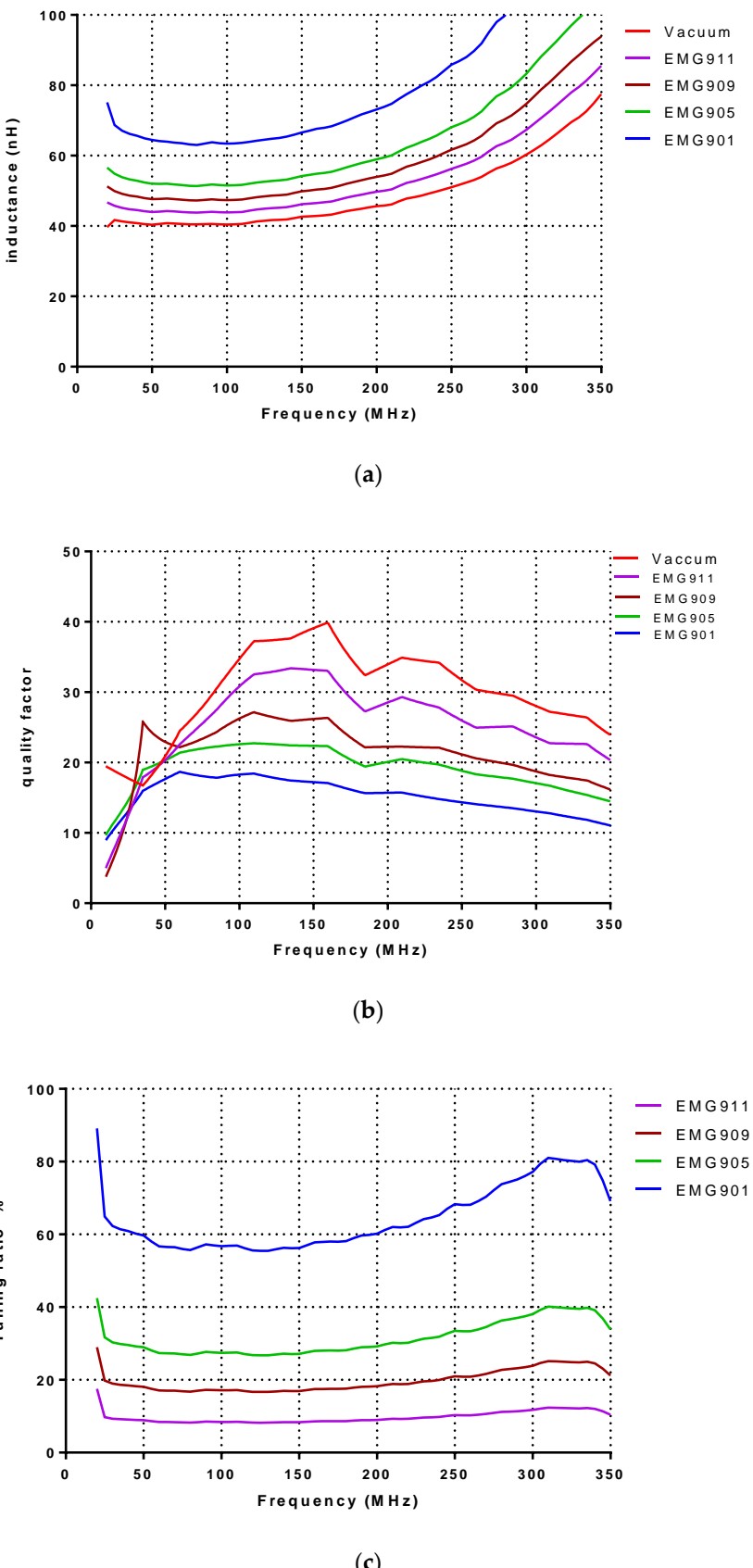

**Figure 8.** Measurement results of wire bond inductor. (**a**) Measured inductance, (**b**) Quality factor, and (**c**) Tuning ratio.

**Table 4.** Characterization of wire-bonded structure tunable inductor.

| Wire-bonded @150MHz | Sim. L (nH) | Sim. Q | Meas. L (nH) | Meas. Q | Sim. TR% | Meas. TR% |
|---|---|---|---|---|---|---|
| Vacuum | 46.5 | 21.52 | 42.6 | 39.1 | - | - |
| EMG901 | 160.79 | 49.594 | 66.6 | 17.2 | 245.7% | 56.2% |
| EMG905 | 100.77 | 44.177 | 54.2 | 22.3 | 116.7% | 27.2% |
| EMG909 | 81.19 | 36.067 | 49.8 | 26.2 | 74.6% | 16.9% |
| EMG911 | 59.36 | 26.498 | 46.2 | 33.2 | 27.6% | 8.45% |

*4.3. Comparison Fabricated Inductor to Previous Work*

A comparison table to compare the proposed techniques, wire-bonded and wire-wound inductors with two other solenoid inductors is provided in Table 5. The wire-bonded and wire-wound inductors show a maximum tuning ratio of 81% and 90.6% at 310- and 300 MHz with EMG901, respectively, which is quite a bit higher than the one achieved by (Yoon et al. 1999b) at 5 MHz, but comparable to the one achieved by (Ning et al. 2006) at 100 MHz. However, the quality factor of the proposed inductor (29.88 and 45.2 at 310 and 300 MHz) are much higher than the one measured by (Vroubel et al. 2004). This shows that the proposed techniques maintain a high quality factor, which is necessary for RF applications.

The measured peak quality factor for wire-bond and wire-wound solenoid inductors are 18.4 at 108 MHz and 69 at 30 MHz, respectively, when EMG901 is fully injected. However, depending on the application and the desired tuning range, the quality factor can be increased in lower-permeability EMG liquids are selected. In this case, a lower tuning ratio is achieved.

**Table 5.** Characterization of wire-bonded structure tunable inductor.

| Parameters | Wire Bond Inductor | Wire Wound Inductor | (Vroubel et al. 2004) [27] | (Ning et al. 2006) [28] |
|---|---|---|---|---|
| Type | Solenoid | Solenoid | Solenoid | Solenoid |
| Technique | Liquid-based core | Liquid-based core | Magnetic core tuned by insulated core | thin-film ferromagnetic (FM) core |
| Maximum tuning ratio | 81% | 90.6% | 18% | 85% |
| Quality factor | 12.7 | 31.3 | 5 | <2 |
| Peak quality factor | 18.4 @108MHz EMG901 | 69 @30MHz EMG901 | 18 | 2 |
| Frequency | 310 MHz | 300 MHz | 5 MHz | 100MHz |

## 5. Conclusions

The tunability of two microfluidic inductor structures were investigated; wire-wound inductor and wire bond inductor. Oil-based ferrofluids as high permeable liquids were used as microfluid. The liquid is being injected through microchannel created for bot inductor structure to alter its permeability property, thus giving a tunable inductance. For the wire-wound structure, the highest tuning range was achieved with EMG901 ferrofluid equal to 90.6% at 300MHz, while the best quality factor was maintained. The wire-bonded structure, which has a high potential to use in the MEMS process, achieved a tuning ratio of 81% at 310 MHz with EMG901 ferrofluid. As for a conclusion, a tunable inductor with high quality factor can be realized using microfluidic techniques.

**Author Contributions:** Several authors were involved in this project. F.B.T.H. was former PhD student in this project. She involved from conceptualization, methodology and data curation including original draft writing. N.A.R., S.S.H. and K.A.M.Z. were research officer in this project. They involved to assist F.B.T.H. in validation method, formal analysis and data curation. M.T.M. is co-supervisor PhD to F.B.T.H. He involved on validation the measurement method, review and editing. The corresponding author is main supervisor to F.B.T.H. He involved in conceptualization, visualization, investigation, formal analysis, original draft writing and project administration. S.K.K.M.: Formal analysis, Writing-review & editing. A.A.M.: Conceptualization, Supervision, Writing-original draft, Writing-review & editing. All authors have read and agreed to the published version of the manuscript.

**Funding:** This research was funded by research university (RU) grant (Universiti Sains Malaysia) with grant number 1001/PELECT/8014010 and Universiti Sains Malaysia (USM) bridging grant with grant number 304/PCEDEC/6316088.

**Acknowledgments:** The author would like to thank the Universiti Sains Malaysia and Collaborative Microelectronic Design Excellence Center (CEDEC) for all the research facilities provided for this project.

**Conflicts of Interest:** The authors declare no conflict of interest. The funders had no role in the design of the study; in the collection; analyses, or interpretation of data; in the writing of the manuscript, or in the decision to publish the results.

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
