# Peer review of "Fabrication and Characterization of Ferrofluidic-Based Wire-Wound and Wire-Bonded Type Inductor for Continuous RF Tunable Inductor"

_applsci, doi:10.3390/app10113776_

Round 1
Reviewer 1 Report
The authors present a study on a fluidic-core ferromagnetic solenoid inductors with tunable inductance for RF applications. An implementations of wire-wound and wire-bond solenoid inductors are presented. By injecting four ferromagnetic liquids with different permeability through the micro-pipe core, the authors were able to achieve varied inductance and a wide range of tunable inductance ratio.
Please have a look at my comments below in four sections:
Graphical illustrations and graphs
- Figure 1 and 2: notations on inductors’ dimensions, scale bars,
- Figure 2a, if it is possible please provide a side-view image of the wire-bond inductor. Because wire bonding normally has a semi-circle shape wires and in Figure 3, the 3D view of wire-bond inductor looks like a triangular shape.
- Figure 4 and 5, 6 and 7: one could make the same scale of y-axis in Figure 4 and 5, 6 and 7 for inductance, quality factor for a better comparison.
- Recommended references to be added:
- The paper has some overlapping results with the paper that has not been cited. “A Novel Ferrofluids-Based High-Tuning High-Q Variable Solenoid Inductor for Frequency-Reconfigurable Circuits Applications, Fatemeh et al. 2015.”
- The idea of wire-bond inductors with Ferrofluid-Based core “A Design of Ferrofluid-Based Fine-Tunable Solenoid MEMS Inductor Using Wire Bonding Technique for Ultra-High Frequency Applications, Ahmad et al. 2019.”
Experimental setup
- More details on how the experiments were conducted would be needed.
- Please provide a more detailed description of the ferrofluids that were used.
- Please specify the pipe specifications: material, inner diameter of wire-wound inductors
- In section 3.1 about wire-wound inductor. It is written that fluids were injected through a pipe by two pipettes as shown in Fig. 1b, however it is not clear the injection parameters. What is the state of the fluids in the pipe after injection? Does it stay steady after injection, or with a flow at a constant flow rate, or with a varied flow rate?
- Please clarify if the presented experiments were conducted at a specific flow rate of ferrofluids in the pipe.
If it was a steady fluid after injection meaning that the particles stay steady, the inductors essentially would have a composite ferromagnetic core, i.e. a mixer of ferromagnetic particles in a non-conducting medium. Inductors with composite core can achieve higher core concentration that yields high core permeability. The same comment applies for wire-bond inductors because inductors are immersed in a microchannel that is technically a steady reservoir.
Please comment on the difference between inductors with composite core and the proposed inductors using ferrofluids as core material.
Please also comment on the particle-distribution uniformity of the ferrofluids under the effect of gravity.
Results and discussions:
- Simulation model: Please describe how the ferrofluid was modelled? Is the ferrofluid considered as a solid magnetic bar with a specific permeability? How is the permeability modelled? In other words, one could characterize the ferrofluids themselves to acquire frequency-dependent permeabilities to import to the model.
- Measurement setup: Please describe the measurement setup that was used to characterize the fabricated inductors, for example which machine was used for measurement and what is the calibration procedure?
- Simulated and measured results:
- Inductance: Simulated and measured inductances should be compared at the same frequency for the same inductors. Please comment on this in section 3.1 and 3.2.
- Inductance difference: there is a large mismatch between simulated and measured inductance of wire-wound inductors and wire-bond inductors. More specific, wire-wound inductors with EMG901 core has Lsimulated = 187.3 nH at 100 Mhz, Lmeasured = 101 nH at 200 MHz. Wire-bond inductors with EMG901 core has Lsimulated =286.3 MHz at 300 MHz, Lmeasured = 66.6 nH at 150 MHz and above 100 nH at 300 MHz. Giving differences in tuning ratio between Figures 4c and 5c, and Figures 6c and 7c. Please comment on the accuracy of the model and measurement error bar.
- Resonant frequency. There is a difference in resonant frequency of the inductors from simulation (Figure 4a) and measurement (Figure 5a). In simulations, the resonant frequencies vary from about 300 MHz to 1300 MHz for inductors with EMG901 and vacuum as core medium. However, the measured frequencies are from approximately 500 MHz to 600 MHz. The same trend is observed also from wire-bond inductor. The frequencies are lower in measurements than in simulation. For example, measured Qpeak of wire-bond inductors are two-fold lower than the simulated values. Please comment on the model of parasitic capacitance and how was parasitic capacitance calibrated? How does parasitic capacitance change with four ferrofluids?
- Please also comment on the difference in parasitic capacitance between wire-wound and wire-bond inductors. Because in wire-wound inductors, ferrofluids are enclosed in the pipe while wire-bond inductors are immersed in a microchannel meaning that there is ferrofluid outside of the solenoid wires. A simple experiment to compare the two cases would be to immerse the wire-wound inductor in a mm-scale channel.
- Implementation for practical applications: with wire-wound inductors and wire-bond inductors in RF circuits and for what applications? Please comment on the implementation for practical applications.
Additional information/data
- Inductance/tuning ratio versus testing condition of ferrofluids such as flow rate: it would be nice also to investigate how inductance would change at different testing condition of ferrofluids. One simple parameter could be the flow rate of ferrofluids in the magnetic core pipe of wire-wound inductors.
- Please also comment on the practical use of the proposed inductors for RF applications. Because of the need of a microfluidic channel, could you suggest how would such inductors be implemented in RF circuitry?
Author Response
The correction and feedback for reviewer is at attachment.
Please see the attachment

Reviewer 2 Report
- In introduction section a table with the state of arts in discrete inductor containing tuning range, quality factor… achieved would facilitate the interest of the present results.
- Pag 80- micropipe” and Pag 82 “micro-pipe”. Unify spelling
- 128-130: In order to clarify, make a table with all the experimental and simulated values.
“four ferrofluids, referred as EMG901, EMG905, EMG909 and EMG911, were applied to, respectively, initial magnetic permeability of 5.4, 3.1, 1.9 and 1.3 magnetic saturation equal to 66mT, 44mT, 22mT and 11mT, and magnetic particle concentration 11.8%, 7.8%, 3.9% and 2%.”
The same applies to “3.2 Wire-bonded solenoid inductor” case.
- The article lacks a section dedicated to the discussion of the results obtained. The authors should comment on the discrepancies between the theoretical simulations and the experimental results of the two types of wire inductors (Figures 4 compared to Figures 5 and Figures 6 compared to Figures 7). Comments related to the factors causing these discrepancies in inductance values, the position of the maximum, the quality factor and de tuning ratio should be added.
- 170-171 : Figs 11 (b) and 11(c) are 1(b) and 1(c)
Round 2
Reviewer 2 Report
Accept in present form
Author Response
Thanks for review my papers, here my comment for reviewer 2:
As referring at section comments and suggestions to authors, reviewer 2 give comment as "accept in present form",thus, the point for reviewer 2 already answer in 1st reply.
Thanks